# Quantitative Assessment of Breast-Tumor Stiffness Using Shear-Wave Elastography Histograms

**DOI:** 10.3390/diagnostics12123140

**Published:** 2022-12-13

**Authors:** Ismini Papageorgiou, Nektarios A. Valous, Stathis Hadjidemetriou, Ulf Teichgräber, Ansgar Malich

**Affiliations:** 1Institute of Diagnostic and Interventional Radiology, Jena University Hospital—Friedrich Schiller University Jena, Am Klinikum 1, 07747 Jena, Germany; 2Institute of Radiology, Suedharz Hospital Nordhausen, Dr.-Robert-Koch-Str. 39, 99734 Nordhausen, Germany; 3Applied Tumor Immunity Clinical Cooperation Unit, National Center for Tumor Diseases (NCT), German Cancer Research Center (DKFZ), Im Neuenheimer Feld 460, 69120 Heidelberg, Germany; 4Center for Quantitative Analysis of Molecular and Cellular Biosystems (Bioquant), Heidelberg University, Im Neuenheimer Feld 267, 69120 Heidelberg, Germany; 5Applied Computer Science, Cyprus International Institute of Management, Akadimias Avenue 21, 2107 Nicosia, Cyprus; 6Department of Biological Sciences, University of Cyprus, 1678 Nicosia, Cyprus

**Keywords:** elastography, RGB histogram, classification, image preprocessing, ultrasound, breast cancer, data curation

## Abstract

**Purpose:** Shear-wave elastography (SWE) measures tissue elasticity using ultrasound waves. This study proposes a histogram-based SWE analysis to improve breast malignancy detection. **Methods:** N = 22/32 (patients/tumors) benign and n = 51/64 malignant breast tumors with histological ground truth. Colored SWE heatmaps were adjusted to a 0–180 kPa scale. Normalized, 250-binned RGB histograms were used as image descriptors based on skewness and area under curve (AUC). The histogram method was compared to conventional SWE metrics, such as (1) the qualitative 5-point scale classification and (2) average stiffness (SWEavg)/maximal tumor stiffness (SWEmax) within the tumor B-mode boundaries. **Results:** The SWEavg and SWEmax did not discriminate malignant lesions in this database, *p* > 0.05, rank sum test. RGB histograms, however, differed between malignant and benign tumors, *p* < 0.001, Kolmogorov–Smirnoff test. The AUC analysis of histograms revealed the reduction of soft-tissue components as a significant SWE biomarker (*p* = 0.03, rank sum). The diagnostic accuracy of the suggested method is still low (Se = 0.30 for Se = 0.90) and a subject for improvement in future studies. **Conclusions:** Histogram-based SWE quantitation improved the diagnostic accuracy for malignancy compared to conventional average SWE metrics. The sensitivity is a subject for improvement in future studies.

## 1. Introduction

Cancer tissue is “stiffer” than normal breast tissue. The stiffening process, a result of cell proliferation and extracellular matrix alterations, was correlated with cancer biology, perhaps with metastatic potential [1,2]. Shear-wave elastography (SWE™) is a sonographic method for the detection of tissue stiffness by measuring the attenuation of a shear wave delivered perpendicular to the tissue [3,4,5,6] (Figure 1a). Several SWE tissue applications [4,7] in the muscle–skeletal system [8] and liver [9,10,11], as well as innovative applications in vessel imaging [12], the small intestine in Crohn’s disease [13], thyroid nodules [14,15], the prostate [16] and breast ultrasound [17], ought to improve the diagnostic yield of B-mode. SWE implements a technology of two crossing ultrasound waves to gain an unbiased and user-independent estimate of tissue elasticity, which is considered advantageous compared to the mechanical pressure-based strain-elastography (SE) method [18]. The Guidelines and Recommendations of the World Federation of Ultrasound in Medicine and Biology (WFUMB) define the standard parameters for all clinically approved elastography applications, including those for SWE and SE in breast imaging [9]. SWE is a promising emerging asset to breast cancer screening, with a sensitivity (Se) and specificity (Spe) of ca. 96/85% [19]. SWE maps, though user-independent, are still vendor-specific and not directly comparable among devices—a hurdle for assembling large databases with high training quality for artificial intelligence (AI). This study used SWE images acquired with an AixPlorer^®^ setup (SuperSonic Imagine™, Aix-en-Provence, France) in a single breast-cancer screening unit.

In contrast to qualitative metrics for strain elastography, known as the 5-tier Tsukuba–Ueno scale [20], SWE allows for a clear parametric readout of tissue stiffness in kPa. To effectively join function with anatomy, manufacturers project the kPa color-adjusted heatmap over the B-mode image to create a user-friendly joint image called an “elastogram” (Figure 1b). The SWE interface supports the custom definition of regions of interest (ROI) and calculates the average and maximum stiffness as characteristic tumor parameters (Figure 1b and Figure 2a). It is, however, a user´s interpretation as to whether the ROI encompasses or partially segments the tumor margins in B-mode. Hence, despite the clear parametric readout, two potential bias parameters emerge; (i) the user-defined ROI selection and (ii) the location of maximum attenuation of the shear wave. Previous reports show that the maximum shear wave attenuation sometimes occurs in the tumor’s surroundings [19], known as the “stiff rim” feature [21,22] (Figure 2a), which the strictly tumor-encompassing ROI definition might neglect. This study aimed to introduce an unbiased, quantitative, user-independent interpretation method of elastograms by extracting the red-green-blue (RGB) histograms. We conclude that the suggested histogram evaluation should provide improved, bias-free breast-tumor classification performance compared to the local average SWE metrics. In our future targets, we shall aim to define database reproducibility and standardization as essentials for the qualitative training of AI algorithms. 

## 2. Materials and Methods

### 2.1. Patient Selection Criteria and Ground Truth

The study was retrospective for n = 73 female patients (n = 96 images) with histologically characterized breast tumors (n = 22/32 patients/unique lesions benign and n = 51/64 patients/unique lesions malignant). Patients were randomly recruited from a single-center breast-cancer database using randomization software (https://www.randomizer.org/, accessed 1 November 2022). An ultrasound-guided biopsy was the single inclusion criterion and served as the ground truth. No exclusion criteria were applied. 

### 2.2. B-Mode and Shear-Wave Elastography Imaging

All patients were investigated with ultrasound in B-mode and SWE in a single session by the same user with more than 15 years of experience in elastography (AM). We used an Aixplorer^®^ ultrasound setup (SuperSonic Imagine, Aix-en-Provence, France) with a SuperLinear™ probe SL 18-5 (5–18 MHz). The colored SWE spectrogram (known as “elastogram”) was the final diagnostic readout, displayed as an over-projection to the B-mode image in parallel screen mode (Figure 1b and Figure 2a). The 0–180 kPa heatmap color scale is a non-adjustable manufacturer´s standard feature, assigning low kPa values (high elasticity and low stiffness) to cool colors and high kPa values to warm colors (high tissue stiffness). A linear correlation is assumed as the baseline for further calculations (Figure 1b). 

### 2.3. Image Analysis for Shear-Wave Elastography 

The workflow for image analysis is graphically summarized in Figure 2. In detail, SWE images were analyzed by (a) a qualitative 5-point risk assessment scale, or the Tsukuba–Ueno scale, as suggested by Itoh et al. for strain elastography [20] and further implemented onto SWE by various groups [23,24,25] (Figure 2b), (b) the local average ROI-based metrics (Figure 2c), and (c) RGB histograms (Figure 2d). 

#### 2.3.1. Local Average SWE Metrics

SWE measures tissue stiffness in kPa. For the measurement, the SWE image is coupled to the B-mode image in a double-screen mode. The measurement region is defined with a circular ROI, which is manually positioned in the center of the target lesion as defined by the B-mode anatomical boundaries. The final readout is (1) the average tumor stiffness (SWEavg) and (2) maximal tumor stiffness (SWEmax) in kPa (Figure 2c). A reference ROI is placed in the healthy surrounding tissue (assigned as “ref”). The ratio of the average stiffness in the target ROI to the reference ROI (surrounding fat pad) is trademarked as the Q-box™ value (Figure 2c). 

#### 2.3.2. RGB Histogram

SWE heatmaps encompassing the target tumor and its surroundings were processed as described in (Figure 3). Heatmaps were orthogonally segmented to .jpeg data (Figure 3a,c) and imported to MATLAB 2017b (The MathWorks, Inc., Natick, MA, USA) using the inbuilt function “imread”. The histogram conversion assumes a linear correlation between (i) kPa and heatmap color code and (ii) during color-to-grayscale transformation. Grayscale components, such as black signal voids, were rejected. After R, G, and B channel extraction, each channel was augmented to fit a 256 × 256 binned matrix and exported to a 250-binned histogram using “imhist” (Figure 3b,d). Cumulative histograms for benign and malignant tumors were compared using the Kolmogorov–Smirnoff test in MATLAB (“kstest2”). Moreover, we quantitatively described the histograms by the area under curve (AUC) and interrogated the discriminative accuracy of the AUC by receiver operating curve (ROC) analysis. 

### 2.4. Statistics and Software 

Statistics and plots were performed in SigmaPlot (Systat Software, Inc., San Jose, CA, USA) and MATLAB 2021b (The MathWorks, Inc., Natick, CA, USA). Data distribution normality was assessed with a Shapiro–Wilk test. Parametric data were compared with a Kruskal–Wallis ANOVA on ranks with Dunn´s post-hoc test or Mann–Whitney rank sum test. Histograms were vectorized and compared with a Kolmogorov–Smirnoff test. ROC analysis was applied in SigmaPlot using an inbuilt macro. Graphical processing and halftones were supported by Inkscape (License name: GPL v2+, https://inkscape.org, accessed 1 October 2022). 

## 3. Result

From the elastograms of n = 96 breast tumors (one image per tumor), we extracted RGB histograms with information from the tumor and its surroundings. Warm color (red) pixel values represent the stiff tumor components; blue pixel values stand for soft or fat-containing elements, whereas the green (“intermediate”) color is assigned to the tissue of intermediate stiffness (Figure 2a,d). Examples of RGB histogram extraction from a benign and malignant lesion are provided in Figure 3. 

### 3.1. RGB Histograms Discriminate Malignant from Benign Tumors

Figure 4 illustrates the cumulative histograms for each channel (Figure 4a, warm colors; Figure 4b, intermediate colors; Figure 4c, cool colors) and tumor identity (gray line for benign, n = 32 and black line for malignant, n = 64). The Kolmogorov–Smirnoff statistical test showed a significant difference between malignant and benign lesions for all color channels, especially for the warm-color (*p* = 5.5 × 10^−8^) and cool-color (*p* = 1.36 × 10^−4^) components. The observed peak-broadening and rightward shift of the warm-color channel (Figure 4a), as well as the corresponding left-sided shift of the cool color channel (Figure 4c), graphically represent the increase in hard-tissue components (high-intensity bins, warm colors) and decrease in soft-tissue elements (low-intensity bins, cool colors) in cancer-including SWE image batches. We hypothesized that, besides the peak and average value descriptors, histograms include information about tissue anisotropy and texture that can enhance the discriminative capacity of SWE as a standalone tool [26,27]. 

### 3.2. The Reduction of Soft-Tissue Components Is a Potential Biomarker for Breast Malignancy 

We further quantified RGB histograms by calculating the integral tissue signal for cool and warm colors, expressed as the AUC. To increase the discrimination capacity, the histograms of cool and warm colors were divided into “low-intensity bins” and “high-intensity bins.” From a whole histogram range of 1–250 bins, we rejected bins 0–14 and 201–250 as noise and defined the bin ranges 15–100 and 101–200 as “low” and “high-intensity bins, ” respectively. A rank sum comparison between benign and malignant lesions (Figure 4d–g) revealed a significant low-bin increase of cool colors in malignancy (Figure 4f); otherwise there were no significant differences (Figure 4d,e,g). Increase in low-intensity cool-colored bins is interpreted as a reduction of soft-tissue components in breast cancer compared to benign tumors. Interestingly, warm colors representing stiff tumor components showed no significant variation between cancer and benign tumors in this dataset (Figure 4d,e). 

Hence, in this dataset, the reduction of soft-tissue components was more sensitive to discriminate malignancy than the increase in hard-tissue elements. 

### 3.3. Receiver Operating Curves for RGB Histograms

Parametric testing of the AUC highlighted the low-intensity cool-colored bins as a potential cancer SWE biomarker with *p* = 0.03 at 95% CI (Figure 4f and Figure 5a). ROC analysis shows a low diagnostic accuracy, with sensitivity = 0.35 for specificity = 0.90. Increasing the sensitivity to 0.75 reduces the specificity to 0.30 (Figure 5b). 

In contrast to the histogram analysis, local average SWE metrics in tumor-bound ROIs failed to reveal a statistically significant difference for malignancy in the same database (Figure 6a–d). Benign tumors were examined as a total and as dignity subgroups, fibroadenoma (FA), scar, in situ carcinoma (CIS), and adenosis (A). The average tumor stiffness (Figure 6a) and the maximum measured tumor stiffness (Figure 6b) showed a high variance (dots outside the box plots show the 75th to the 95th percentile) but were not significantly increased in cancerous lesions (histological ground truth) compared to lesions of benign nature, *p* > 0.05 Kruskal–Wallis ANOVA on ranks. 

All in all, standalone analysis of SWE signal as implemented in this study discriminated between malignant and benign lesions (Figure 5) better than conventional descriptors (Figure 6), although still with a low diagnostic accuracy, which needs to be improved in future studies. 

One of the technical drawbacks of the suggested RGB analysis method is the logistic demand and the need for integrated software with standard reference curves. The widely implemented Tsukuba 5-tier scale for the risk stratification of solid tumors in clinical practice requires no additional software. The Tsukuba scale tumor grading is only qualitative and empirical, based on the color appearance of the SWE image as graphically shown in Figure 7a. The color map is vendor-defined and cannot be thresholded by the user (unlike the doppler signal), therefore is a reliable and reproducible correlation between tissue stiffness in kPa and color representation. Still, it is inevitably subjected to user´s expectation based on the B-mode image. For this research dataset, the Tsukuba breast cancer (CA) score was statistically higher compared to the benign categories, *p* = 0.015 Kruskal–Wallis ANOVA on ranks (Figure 7). Nevertheless, the qualitative elastogram grading was not blinded to the B-mode features, which is, as commented above, a very likely source of bias. 

Our results show that the quantitative heat map evaluation using RGB histograms could independently influence the diagnostic outcome. 

## 4. Discussion

In this research project, we exploited the discriminative potential of the quantitative histogram analysis in breast SWE. In a dataset where the local average and peak quantitative values failed to classify malignancy (ANOVA *p* > 0.05, Figure 6), the split RGB histogram-based approach significantly improved the sensitivity of SWE for malignant versus benign lesions with *p* < 0.0001 at the 95% confidence interval. ROC analysis defined the reduction of soft-tissue components as a promising biomarker (Figure 5). Future studies should be designed to prove the current results, and improve the diagnostic accuracy by including clinical data along information with intra- and peritumoral regions such as in previous prominent research [28]. 

### 4.1. Discriminative Value of the Local SWE Average, State of the Art 

Previous studies are mainly based on analyzing average and maximum SWE values from within the tumor boundaries. Au et al. [29], in a series of N = 123 masses, concluded that the elasticity ratio, i.e., the average strain ratio of the tumor to its surroundings (a Q-box equivalent), is the most sensitive discriminative factor for malignancy amongst strain descriptors. Yoon et al. [30] studied a series of N = 267 breast masses in order to highlight the value of the maximum SWE within tumor boundaries as the optimal discriminative criterion for malignancy. Yang et al. [31] used supersonic elastography in a study encompassing N = 226 subjects and showed a high discriminatory power of the mean and SD SWE values. In the latest peer-reviewed meta-analysis, Pillai et al. [32] included N = 19043 patients in a single receiver operating curve, proving the average SWE within tumor boundaries as the most sensitive discriminative parameter with an area under the curve (AUC) = 0.93. 

### 4.2. The Role of Anisotropy in Tumor Classification

All above-cited studies focus on SWE measurements acquired within the tumor boundaries; the tumor surroundings serve as reference tissue for calculating an elasticity ratio [29]. However, amid previous reports on the high discriminative power of the average SWE, maximum SWE, or Q-box, in our medium-scale database of N = 96 breast masses, the standard SWE scores failed to discriminate for malignancy (Figure 6). Interestingly, the qualitative color map scale outperformed the SWE metrics and could differentiate between malignancy and benign tumors with a statistical significance illustrated in Figure 7. Evidence supporting the visual color map assessment of SWE elastograms has its foundations in numerous previous studies [23,24,30]. As noted above, the visual evaluation objectivity is jeopardized by B-mode-injected bias, which could explain the Ueno-scale superiority versus SWE values in this dataset. However, color maps include information on tumor anisotropy, which can add to their informational value. In a previous meta-analysis by Blank et al. [33] a series of N = 2989 patients revealed the importance of “stiffness gradient,” i.e., the slope between the maximum and minimum SWE tumor value, as an essential discriminative factor that outperformed standard measurements by 28% in malignancy classification. Slope gradient is a mesoscopic approach to quantify tumor anisotropy, further supported by evidence from the groups of Chen et al. [26] and Zhang et al. [27]. 

### 4.3. Histogram SWE Analysis, Experience from Previous Studies, and the Role of Deep Learning 

In a previous study, Carlsen et al. reported equal discriminative efficacy for histograms and elastographic ratios [34] in a study paradigm implementing strain elastography (in this study, we implemented shear-wave elastography). Moreover, the study design of Carlsen et al. [34] focused on BIRADS-3 cases, whereas our research encompasses all BIRADS classes. Xue et al., in a more extensive study of N = 500, confirmed that the Ueno scale method performed equally to the SWE average values [35]. 

The approach of crossing the borders of qualitative color maps and extracting RGB histograms from the SWE data appear in previous literature, such as by Cheng et al. [36]. The authors used liver examinations to extract RGB histograms from color maps and merge them to organ “ultrasomics.” Remarkable achievements in breast-tumor classification were published by Zhang et al., Zhou et al., and Jiang et al. [37,38,39]. Independent researcher groups agree that convolutional neuronal network (CNN) training with cumulative B-mode and SWE data boosted the discriminative efficiency for malignancy. Part of the deep learning (DL) process implemented to improve the classification relies on the extraction of RGB histograms.

The essentiality of DL transparency, recently promoted under the term “explainable AI,” re-enforces the importance of knowing the independent effect of the intermediate processing steps. Notably, the suggested approach is free of hidden meta-data that may draw the attention of AI systems, such as numbers, letters, or other non-diagnostic image features. Such features can bias the results of sophisticated explainable AI (XAI) algorithms, meticulously summarized by the group of Van der Velden [40]. XAI training databases might benefit from data curation following the paradigm of this manuscript. 

### 4.4. Effect of the ROI Size and Surrounding Tissue in SWE Interpretation

A final point that the authors would like to comment upon is the effect of ROI selection, size, and boundaries on the final SWE readout. Previous studies [41] have elucidated that the ROI size is an independent modulator of the SWE readout. The ROI-independent quantitative approach introduces objectivity in a traditional user-dependent method in ultrasonography and strips the final judgment from the inevitable B-mode-injected operator bias. 

Previous research [42] implemented segmentation paradigms based on the B-mode intersection border, thus excluding the peritumoral shear-wave alterations. Lee et al., however, increased the accuracy of cancer detection by including the peritumoral tissue, thus elucidating the increased value of co-interpretation of tumoral and peritumoral patterns [43]. The “stiff-rim” sign in SWE describing a ring-like sink in elasticity in the peritumoral tissue was reported by the group of Zhou [22] and by Xu et al. [21]. Zhou et al. [38] introduced a segmentation method for preserving the peritumoral tissue that showed superior performance compared to its tumor intersection border predecessors. Using supersonic SWE, an alternative rim segmentation approach was suggested by Yu et al., implementing a hydraulic model for automatic tumor rim discrimination [31]. Though promising, the method is not yet commercialized, probably due to its high computational requirements. All in all, the increased discriminative capability of the RGB histograms relies, at least partly, on information from the tumoral and peritumoral milieu. In contrast, the average SWE metrics derive from strictly intratumoral selected ROIs. 

### 4.5. Study Weaknesses 

Our study was limited to a low-N, single-center, retrospective observational format. Regarding data interpretation, we would like to remark on the arbitrary use of the Tsukuba scale for SWE—a scale originally established for another elastography method (strain elastography) which may not apply to SWE with the same accuracy. However, SWE is a newly established elastography approach, and the Tsukuba–Ueno scale is the commonly used clinical standard, extrapolated from the classical strain elastography field. Moreover, our system is based on the assumption of a linear correlation between RGB tones and kPa values on the elastograms, which may inflict deviations from reality. Whether AixplorerTM uses a linear or retina-adapted color code is nevertheless a rigid feature, with possible influence on absolute values but not on the intergroup differences in our dataset. In future studies, we opt to address some underexplored points of this work, such as the effect of the lesion size and the definition of cut-off limits for the histogram implementation. Recruiting more patients from the central database and powering a retrospective dichotomized design might prove histograms as problem solvers in ambivalent tumor cases between surveillance and biopsy (BIRADS-3 class). Finally, an exciting point requiring a larger study population is the effect of the lesion size on the accuracy of SWE and the definition of a minimum necessary cut-off tumor size to guarantee elastogram reliability. 

## 5. Conclusions and Clinical Significance

In this study, we applied quantitative shear-wave elastography using RGB histograms. The procedure is free of user bias, easily customizable, and showed improved discriminative ability between benign and malignant breast tumors compared to the standard manufacturer´s stiffness metrics. The RGB histogram evaluation could improve the standalone diagnostic accuracy of the SWE for breast tumors by 28% based on the ROC analysis (Youden index). Future studies should be designed to prove the current results, and improve the diagnostic accuracy by including clinical data along with information from intra- and peritumoral regions.

## Figures and Tables

**Figure 1 diagnostics-12-03140-f001:**
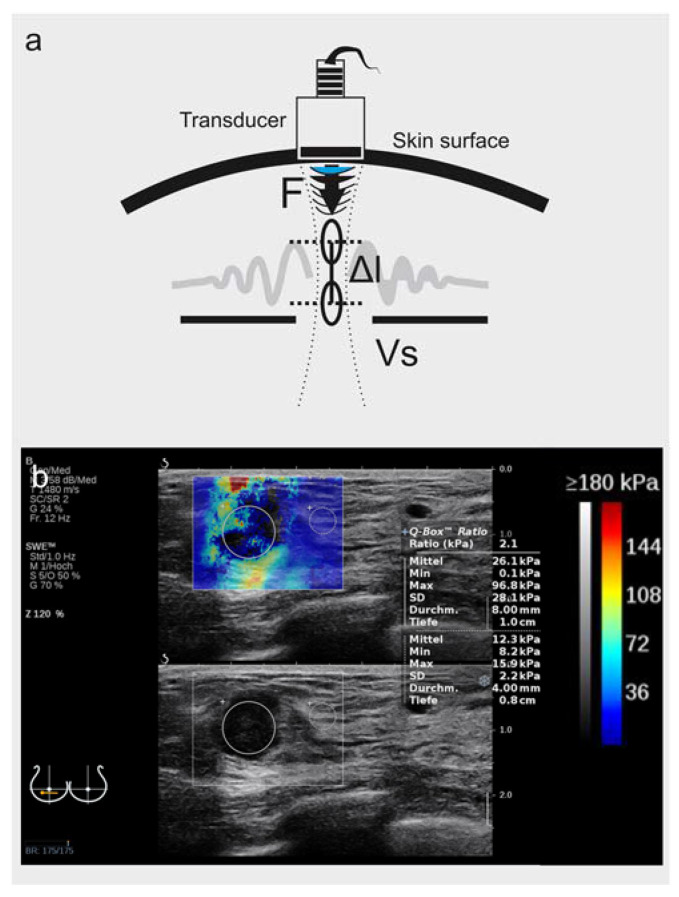
(**a**) Principles of shear-wave elastography. Shear-wave elastography (SWE) measures tissue density or stiffness in kPa through the mechanical distortion caused by an ultrasound wave. The transducer emits a perpendicular acoustic radiation force impulse (F), which causes a focal radial dislocation (Δl) and initiates a transverse wave, known as a shear wave. The propagation speed of the shear wave (Vs) is measured by a detection pulse emitted by the same transducer. The tissue density (ρ) is inversely proportional to the shear wave propagation speed square (ρ ~ 1/Vs^2^). (**b**) Sample image SWE (SuperSonic Imagine). The tissue stiffness of each pixel is measured in kPa and is color-coded in a spectrogram (“elastogram“) which is overlaid to the B-mode image in real time and can be projected in dual-mode to the B-mode image. As a manufacturer’s standard, high stiffness is warm-colored (red), and elastic tissue with low kPa values is cool-colored (blue).

**Figure 2 diagnostics-12-03140-f002:**
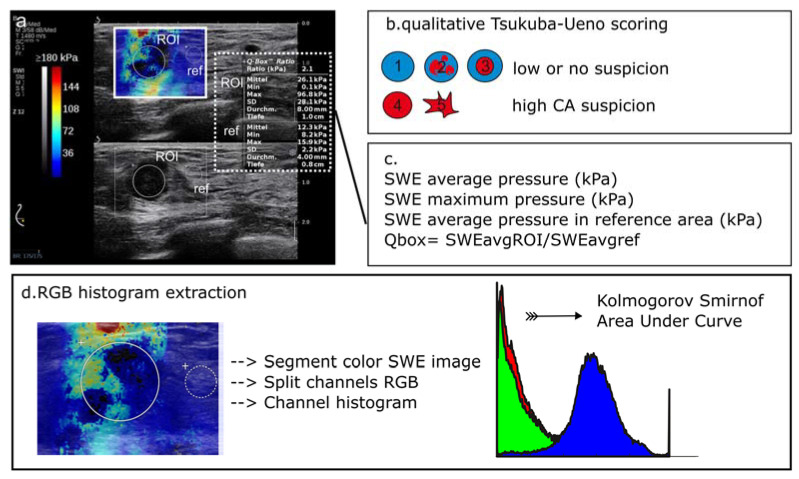
Workflow for image processing. (**a**) Sample image, SWE projected in dual mode with the B-mode parallel imaging. A region of interest (ROI) and a reference region (ref) are defined using the B-mode image to encase the primary lesion (ROI) and a representative reference sample of the surrounding fat tissue (ref). The color map provides pixel-based information on tissue elasticity, explained by the scale bar on the image’s left side and ROI-based statistics on the right (dotted box). Information on the ROI diameter and depth from the skin surface (depth) is also provided. (**b**) Qualitative lesion assessment based on the Tsukuba scale, increasing score from 1 to 5 reflects the probability for malignancy. (**c**) ROI statistics: average, minimum, maximum, and standard deviation (SD) of tissue elasticity in kPa. The Q-box Ratio^TM^ is the fraction of the average target ROI stiffness to the average stiffness of the reference region. (**d**) Segmentation of the whole SWE field for RGB analysis purposes. Image import, RGB channel splitting were performed in MATLAB as described in Materials and Methods section. Histogram differences were compared using the Kolmogorov–Smirnoff test. The diagnostic accuracy of the histogram area under curve (AUC) was interrogated with receiver operating curve (ROC) analysis. CA, cancer; SWE, shear-wave elastography.

**Figure 3 diagnostics-12-03140-f003:**
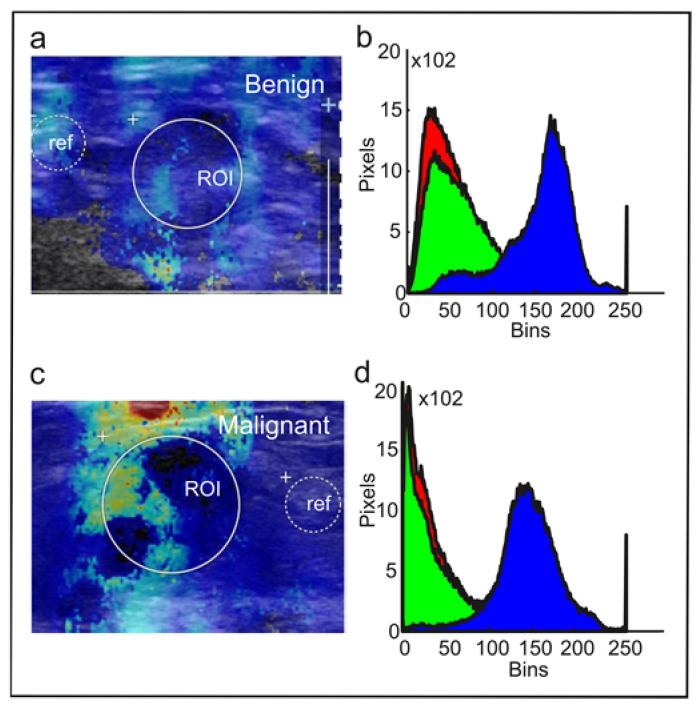
Sample elastogram and histograms. (**a**) Sample image of a benign breast lesion, (**b**) RGB histogram of the lesion (**a**), (**c**) sample image of a malignant breast lesion, and (**d**) RGB histogram of the lesion in (**c**). ROI, region of interest; ref, reference region (adjacent fat pad).

**Figure 4 diagnostics-12-03140-f004:**
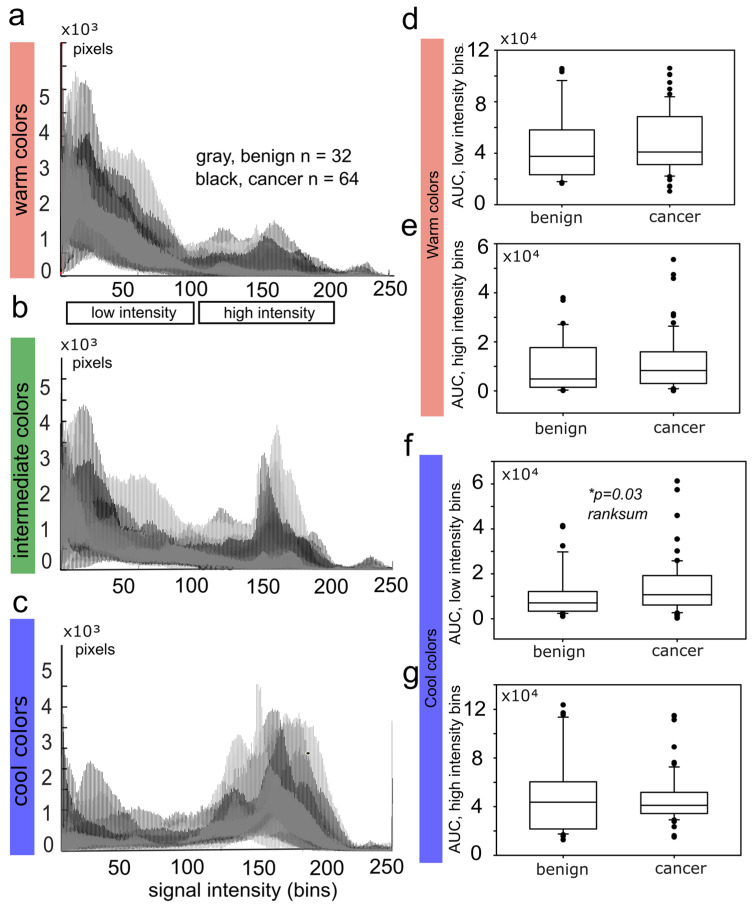
SWE histogram metrics. Cumulative histogram plots for the (**a**) warm colors, *p* = 5.5 × 10^−8^, Kolmogorov–Smirnoff test, (**b**) intermediate colors, *p* = 0.0258, Kolmogorov–Smirnoff test, and (**c**) cool colors, *p* = 1.36 × 10^−4^, Kolmogorov–Smirnoff test. (**d**–**g**) Area under curve (AUC) was calculated for the low-intensity bins (15–100) and the high-intensity bins (100–200). Normality was rejected with Shapiro–Wilk test. The AUC of benign to malignant tumor was compared with Mann–Whitney rank sum test. (**f**) Statistical significance was proven for the low-intensity bins of the cool colors, reflecting a reduction of soft-tissue components, *p* = 0.03, benign n = 32; cancer n = 64.

**Figure 5 diagnostics-12-03140-f005:**
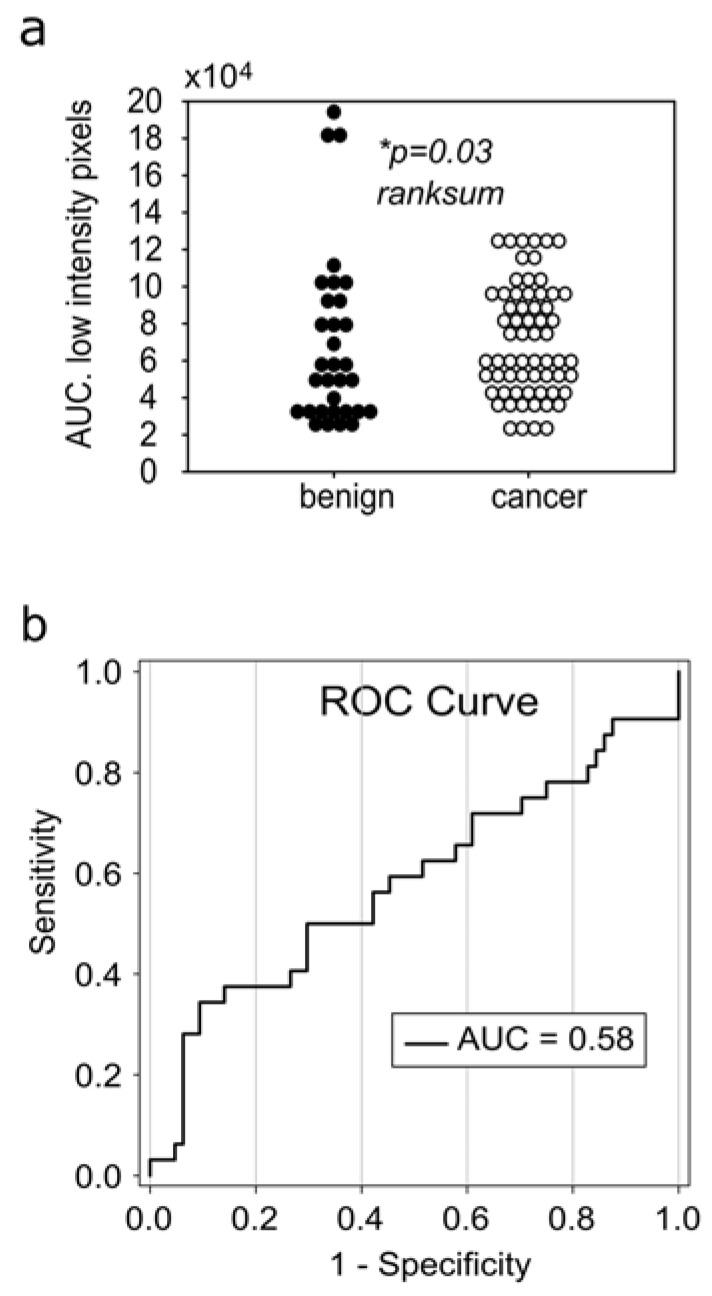
Receiver operating curves (ROC) of cool color histograms. (**a**) The best diagnostic accuracy was revealed for the decreased soft-tissue components (expressed as increased low intensity histogram bins) in malignant lesions, *p* = 0.03, rank sum test. (**b**) ROC shows a low diagnostic accuracy, Se = 0.3/Spe = 0.9, which is not sufficient as a standalone tool, but sheds light onto new image analysis aspects.

**Figure 6 diagnostics-12-03140-f006:**
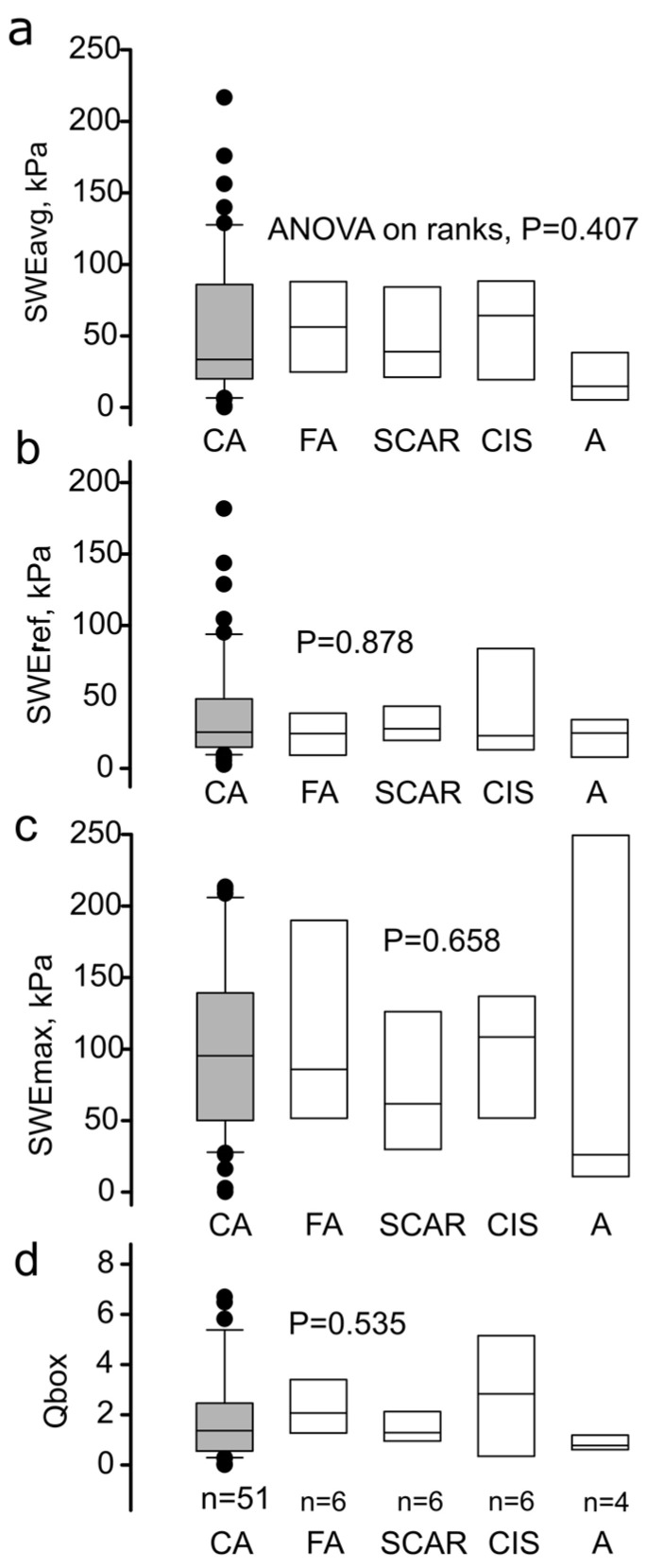
Average SWE metrics of the inbound tumor ROI. (**a**) Average stiffness from target ROI within the B-mode tumor boundaries, (**b**) average stiffness from reference surrounding fat tissue region, (**c**) maximum stiffness from target ROI, and (**d**) Q-box ratio for cancer (CA, n = 62 patients) and benign lesions (n = 32 patients). Box plots show the median and interquartile range, dots within the 5–95th percentile. FA, fibroadenoma; SCAR, scar; CIS, cancer in situ; A, adenosis. (**a**–**d**) *p* > 0.05, Kruskal–Wallis ANOVA on ranks.

**Figure 7 diagnostics-12-03140-f007:**
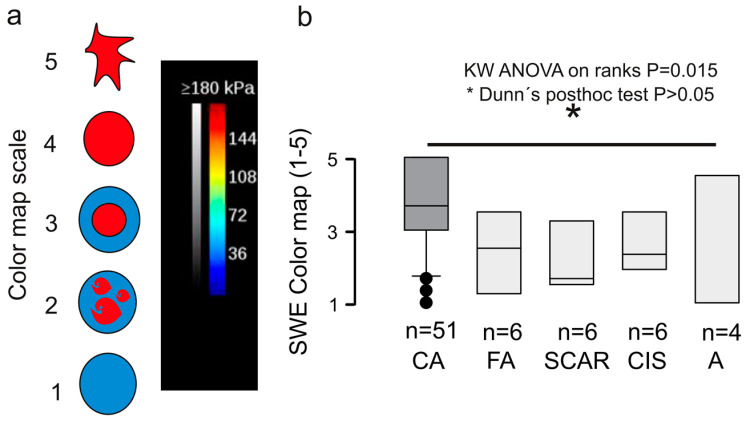
Qualitative elastography assessment. (**a**) Qualitative evaluation of the elastogram in a scale 1–5 with criteria similar to the Tsukuba–Ueno scale (**b**) Box plot showing the median and interquartile range, dots within the 5–95th percentile. Cancer (CA, n = 62 tumors) and benign lesions (n = 32 tumors), *p* = 0.015, Kruskal–Wallis ANOVA on ranks with Dunn’s post-hoc test. Note that the qualitative elastogram grading cannot be blinded to the B-mode features, which is a likely source of bias. FA, fibroadenoma; SCAR, scar; CIS, cancer in situ; A, adenosis.

## Data Availability

Raw data are available, without restriction, in the Appendix A.

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
