# Peer review of "Quantitative Assessment of Breast-Tumor Stiffness Using Shear-Wave Elastography Histograms"

_diagnostics, 2022, doi:10.3390/diagnostics12123140_

Round 1

Reviewer 1 Report

In this study, the authors apply quantitative shear-wave elastography using RGB histograms to improve breast malignancy detection. Histogram-based SWE quantitation showed superior discriminative sensitivity for malignancy compared to tumor-bound SWEavg/SWEmax.

 Some comments about the study:

1. Please provide the sensitivity,specificity and MAP of the histogram-based SWE quantitation based on a certain threshold determined by generally accepted rules.

2. In every case, how many images were selected for analysis? How were the images selected? How was the final binary classification from those images analyzed?

3. Please briefly describe the contrast agent used and the protocol for performing CEUS and for selecting frames to be analyzed in the main text. Was the timing between contrast infusion and image capture standardized? 

Author Response

Summary of changes in the updated manuscript

Dear Editor,

Dear Reviewers,  

Thank you for the time spent on our manuscript corrections and the opportunity to submit the revised version. The comments of our reviewers facilitated a beneficial manuscript restructuring with robust statistical additions. In summary, the revised version has the following upgrades:

  • The terminology “red, green, blue” was changed to “warm, intermediate, and cool colors” to better match the literature.
  • We quantified the histograms based on the AUC, which is a more solid descriptor compared to the curve skewness
  • Reduction of the soft tissue elements was defined as the most sensitive marker for malignancy (figure 4) and used for a ROC-curve analysis (Figure 5)
  • Figure S1 was removed, and the information was integrated into the main text, Figure 4a-c corrected according to the reviewer´s suggestions
  • Figure 4 was updated to include a quantitative analysis of the histogram Area under the curve. The reduction of soft tissue components was proven the most sensitive metric to describe cancer (domination of low-intensity cool color bins), new results subsection “3.2”
  • The average SWE metric was shifted from Fig 4 in the previous version to a new figure 6 and
  • Figure 5 was renamed to Figure 7
  • The IRB number was added (Line 32) after a request from the editor

A point-by-point response to the reviewer´s points is attached. We hope we have clarified all instances and look forward to a fruitful scientific exchange with your peer experts.

CAVE: Kindly note that the line number references correspond to the text version with corrections, and not to the clean manuscript.

With best regards,   

Point-by-point response to reviewer 1

In this study, the authors apply quantitative shear-wave elastography using RGB histograms to improve breast malignancy detection. Histogram-based SWE quantitation showed superior discriminative sensitivity for malignancy compared to tumor-bound SWEavg/SWEmax. Some comments about the study:

R1_C001: Please provide the sensitivity,specificity and MAP of the histogram-based SWE quantitation based on a certain threshold determined by generally accepted rules.

Re: The reviewer raised a fair issue. In the revised manuscript, Figure 4 was updated to include a quantitative analysis of the histogram Area Under Curve, described in the results subsection “3.2”, Lines 252-265. Moreover, a ROC analysis was performed, inserted as new Figure 5, and results in subsection “3.3”, Lines 266-275. ROC showed a low diagnostic accuracy (AUC 0.58, Youden index 1.28) as a more robust method compared to the KS test. ROC analysis, although not sufficient as a standalone method, showed improved results compared to conventional SWE metrics, which could not discriminate between benign and malignant lesions in this dataset.

R1_C002: In every case, how many images were selected for analysis? How were the images selected? How was the final binary classification from those images analyzed?

Re: We thank the reviewer for giving us the opportunity to clarify this point. Text was updated as follows: Lines 160-165: “The study was retrospective for n=73 female patients (n=96 images) with histologically characterized breast tumors (n=22/32 patients/unique lesions benign and n=51/64 patients/unique lesions malignant). Patients were randomly recruited from a single-center breast cancer database using randomization software (https://www.randomizer.org/). An ultrasound-guided biopsy was the single inclusion criterion and served as the ground truth. No exclusion criteria were applied. “

Lines 187-197: “SWE heatmaps encompassing the target tumor and its surroundings were processed as described in (Figure 3). Heatmaps were orthogonally segmented to .jpeg data (Figure 3 a, c) and imported to MATLAB 2017b (The MathWorks, Inc., Natick, MA, USA) using the inbuilt function “imread.” The histogram conversion assumes a linear correlation between (i) kPa and heatmap color code and (ii) during color-to-greyscale transformation. Grayscale components, such as black signal voids, were rejected. After R, G, and B channel extraction, each channel was augmented to fit a 256 x 256 binned matrix and exported to a 250-binned histogram using “imhist” (Figure 3 b, d). Cumulative histograms for benign and malignant tumors were compared using the Kolmogorov-Smirnoff test in MATLAB (“kstest2”). Moreover, we quantitatively described the histograms by the Area Under Curve (AUC) and interrogated the discriminative accuracy of the AUC by Receiver Operating Curve (ROC) analysis.“

R1_C003: Please briefly describe the contrast agent used and the protocol for performing CEUS and for selecting frames to be analyzed in the main text. Was the timing between contrast infusion and image capture standardized? 

Re: The study was based on ultrasound shear-wave elastography images, which are static images (no frame selection was applied), acquired without contrast agent application. 

Point-by-point response to reviewer 2

The paper targets the use of shear wave elastography heatmaps to discriminate between benign and malignant breast cancer. A detailed presentation of the device is followed by the description of the proposed method. A comparison of the standard approach with the proposed 5 point scale represents the main findings, even if a more clear organization of the paper should be addressed. The paper might be suitable for publication in Diagnostics. However, the following requirements have to be fulfilled.

R2_C001: in Figure 2 the ROI is related with a low stiffness region with respect to the red colors above, even if you refer in the caption that it should encase the primary lesion;

Re: We thank the reviewer for requiring this clarification. The ROI boundaries were defined by the B-mode boundaries of the tumor. As the reviewer correctly observed, the “red”, hard SWE zone extends beyond the tumor boundaries. This phenomenon is discussed as the “stiff-rim” sign in the surrounding tissue, Lines 372-390. We rephrased:

L180-181: “ SWE measures tissue stiffness in kPa within circular ROIs, custom-positioned in the center of the target lesion as defined by  B-mode boundaries ”

R2_C002: in Figure 3 the same heatmap is shown, but it is not commented, so you need to explain the two cases (e.g. are they referred to different patients, etc...);

Re: Figure 3 was revised according to the reviewer’s suggestion, and illustrates an example of a benign (Fig 3a) and a malignant (Fig 4a) lesion with the corresponding histograms.

R2_C003: in line 149 find an equivalent way for ”significant significance”;

Re: corrected

R2_C004: in Figure S1 you should adopt the same readable format of Figure 4, showing the average and standard deviation lines;

Re: Figure S1 was integrated into the main figures in the revised manuscript version, as Figure 4 a-c. A unique format was kept, as correctly the reviewer commented.

R2_C005: lines 173-174 have to be explained in a precise manner, because your claims in Figure 4 require a much more detailed and precise description.

Re: We rephrased this essential explanation as follows:

Lines 181-188: “ 2.3.1. Local average SWE metrics: SWE measures tissue stiffness in kPa. For the measurement, the SWE-image is coupled to the B-mode image in a double-screen mode. The measurement region is defined with a circular ROI, which is manually positioned in the center of the target lesion as defined by the B-mode anatomical boundaries. The final readout is (1) the average tumor stiffness (SWEavg) and (2) maximal tumor stiffness (SWEmax) in kPa (Figure 2c). A reference ROI is placed in the healthy surrounding tissue (assigned as “ref”). The ratio of the average stiffness in the target ROI to the reference ROI (surrounding fat pad) is trademarked as the Q-box value (Figure 2c). “

R2_C006: the performance obtained in Figure 5 through the 5-point scale have to be explained with much more details (e.g. how do you choose the thresholds for the colormap?);

Re: Figure 5 was renamed to Figure 7 in the revised version. The Tsukuba scale tumor grading is only qualitative and empirical, based on the color appearance of the SWE-image as graphically shown in Fig 7a. The color map is vendor-defined and cannot be thresholded by the user (unlike the doppler signal), therefore is a reliable and reproducible correlation between tissue stiffness in kPa and color representation.

The main text was updated accordingly, Lines 300-303    

R2_C007. in line 198 what is the sensitivity you are referring to? What is its associated value? Are you referring to a score adopted for classification? Did you use a training and a validation set?

Re: We thank the reviewer for this comment. The manuscript was enriched with a ROC analysis of the AUC, which is a more robust diagnostic accuracy metric than the KS test. In the revised manuscript, Figure 4 was updated to include a quantitative analysis of the histogram Area Under Curve, described in the results subsection “3.2”, Lines 252-265. Moreover, a ROC analysis was performed, inserted as new Figure 5, and results in subsection “3.3”, Lines 266-275. ROC showed a low diagnostic accuracy (AUC 0.58, Youden index 1.28) as a more robust method than the KS test. ROC analysis, although not sufficient as a standalone method, showed improved results compared to conventional SWE metrics, which could not discriminate between benign and malignant lesions in this dataset.

Since we did not apply deep learning, splitting of the dataset to training and validation was not applied.

R2_C008: in lines 215-217 there is the main result of your work, which is not clearly presented previously because your comments of Figure 4 seem to focus on plots with a much less informative content;

Re: In the revised data analysis, the main result is presented in Figure 5. Kindly refer to the R2_C008 response.

R2_C009. In line 225 write explicitly ”by pprox..”;

Re: corrected

R2_C010. in lines 252-270 you should describe the advantages related with the inclusion of clinical data, improving images referred to intra- or peri-tumoral regions, by including the reference • A ultrasound-based radiomic approach to predict the nodal status in clinically negative breast cancer patients, Scientific Reports (2022);

Re: Text addition Lines 325-328: “ Future studies should be designed to prove the current results, and improve the diagnostic accuracy by including clinical data along information with intra- and peritumoral regions such as in previous prominent research [28].”

  1. in Conclusions you should quantify more precisely your ”improved discriminative ability”.

Re: text was updated, Lines 420-422: “ The RGB histogram evaluation could improve the standalone diagnostic accuracy of the SWE for breast tumors by 28% based on the ROC analysis (Youden index). Future studies should be designed to prove the current results, and improve the diagnostic accuracy by including clinical data along information with intra- and peritumoral regions “

Reviewer 2 Report

see report

Author Response

(The authors gave the same response as above.)

Round 2

Reviewer 2 Report

The authors addressed the requirements for publication.